# Advanced Progress in the Role of Adipose-Derived Mesenchymal Stromal/Stem Cells in the Application of Central Nervous System Disorders

**DOI:** 10.3390/pharmaceutics15112637

**Published:** 2023-11-16

**Authors:** Haiyue Wu, Yishu Fan, Mengqi Zhang

**Affiliations:** 1Department of Neurology, Xiangya Hospital, Central South University, Changsha 410008, China; wuhaiyue@csu.edu.cn (H.W.); fanyishu@csu.edu.cn (Y.F.); 2National Clinical Research Center for Geriatric Disorders, Xiangya Hospital, Central South University, Changsha 410008, China

**Keywords:** adipose-derived mesenchymal stromal/stem cells, central nervous system injury, epilepsy, neurodegenerative disease, multiple sclerosis, clinical trial

## Abstract

Currently, adipose-derived mesenchymal stromal/stem cells (ADMSCs) are recognized as a highly promising material for stem cell therapy due to their accessibility and safety. Given the frequently irreversible damage to neural cells associated with CNS disorders, ADMSC-related therapy, which primarily encompasses ADMSC transplantation and injection with exosomes derived from ADMSCs or secretome, has the capability to inhibit inflammatory response and neuronal apoptosis, promote neural regeneration, as well as modulate immune responses, holding potential as a comprehensive approach to treat CNS disorders and improve prognosis. Empirical evidence from both experiments and clinical trials convincingly demonstrates the satisfactory safety and efficacy of ADMSC-related therapies. This review provides a systematic summary of the role of ADMSCs in the treatment of central nervous system (CNS) disorders and explores their therapeutic potential for clinical application. ADMSC-related therapy offers a promising avenue to mitigate damage and enhance neurological function in central nervous system (CNS) disorders. However, further research is necessary to establish the safety and efficacy of clinical ADMSC-based therapy, optimize targeting accuracy, and refine delivery approaches for practical applications.

## 1. Introduction

Stem cells, particularly pluripotent stem cells (PSCs), like induced pluripotent stem cells (iPSCs) and embryonic stem cells (ESCs), have garnered significant attention due to their remarkable capacities for proliferation, pluripotency, self-renewal, and differentiation. Their potential applications in injury repair, tissue regeneration, and targeted differentiation to specific cell types have been extensively explored. However, these therapeutic benefits come with the inherent risk of tumorigenicity, which poses a challenge for their safe clinical application. Addressing this concern, the study of adipose-derived mesenchymal stromal/stem cells (ADMSCs) presents a promising avenue. Classified as a subset of mesenchymal stem cells (MSCs), ADMSCs share regenerative properties with other MSCs. Crucially, as lineage-restricted stem cells, ADMSCs demonstrate a lower risk of tumorigenicity [1,2]. The application of ADMSCs can be achieved through two main approaches: cell transplantation and cell-free therapy using ADMSC-derived exosomes (ADMSC-E) or secretome. Previous studies have demonstrated the multi-directional differentiation capacity of ADMSCs, highlighting their potential in brain damage repair [3,4,5,6]. Under specific conditions, ADMSCs can be induced and transdifferentiated into endoderm, mesoderm, and ectoderm cells, expanding their broad applicability [7,8]. Adipose-derived stem cells (ADMSCs) hold great promise for regenerative therapy, attributed to their subcutaneous accessibility, abundance, and less invasive procurement techniques [9]. Ongoing studies exploring the application of ADMSCs in regenerative therapy stem from its promising potential [10,11,12,13]. In addition to differentiation, like other MSCs, ADMSCs promote wound healing and tissue repair mainly based on paracrine, which modulates the microenvironment comprehensively by secreting a large number of bioactive factors [14]. Therefore, cell-free therapies based on ADMSCs have been studied, mainly consisting of ADMSC-derived exosomes and ADMSC secretome. ADMSC-derived exosomes, which encapsulate regulatory molecules including cytokines, emerges as a promising avenue for treating central nervous system (CNS) diseases [15,16,17]. These exosomes, characterized as single-membrane organelles with diameters ranging from 30 to 200 nm, demonstrate enrichment of selected nucleic acids, proteins, glycoconjugates, and lipids [18]. The secretome is a mixture of various molecules secreted by ADMSCs, including exosomes, microvesicles, cytokines, chemokines, hormones, interleukin, cell adhesion molecules, growth factors, lipid mediators, etc., which regulate the microenvironment by mediating intercellular information transduction through paracrine [19]. Both ADMSC-E and secretome, on the basis of their diverse components, often play important and complex regulatory roles in a variety of biological processes, including development, homeostasis, and immunity, and are involved in neurodegenerative diseases and cancer [18]. For instance, in wound healing, ADMSCs not only secrete numerous growth factors and cytokines but also promote tissue granulation, macrophage recruitment, and vascularization [20,21]. Moreover, ADMSCs exhibit potential for immunomodulation, showing efficacy in the treatment of severe refractory acute graft-versus-host disease, as well as immunological and hematological disorders such as refractory pure red cell aplasia and idiopathic thrombocytopenic purpura [22].

This comprehensive review seeks to elucidate the therapeutic potential of adipose-derived mesenchymal stromal/stem cells (ADMSCs) in addressing a spectrum of central nervous system (CNS) conditions, encompassing CNS injuries, epilepsy, multiple sclerosis, and neurodegenerative diseases. These CNS disorders often involve irreversible neuronal damage. Conventionally, the limited regenerative capacity of mature neurons, the disturbed balance of the microenvironment, along with progressive neuroinflammation and neural destruction have posed challenges to the recovery of the CNS. However, ADMSCs, which have the potential of division and differentiation, as well as the ability of regulation through paracrine, offer a promising pathway for central nervous system restoration. ADMSCs, as a subset of mesenchymal stem cells, possess wide availability, ease of access, and a high safety profile, making them an ideal candidate for brain damage treatment, as evidenced by the extensive research in this field [23,24,25]. Significantly, researchers have achieved successful transdifferentiation of ADMSCs obtained from both human and animal sources, leading to the generation of neurons and glial cells [23,24,25]. In addition, cell-free therapies, including ADMSC-E and ADMSC secretome, utilize the paracrine of ADMSCs to regulate the local microenvironment of nervous system diseases, alleviate neuroinflammation, reduce neural damage, and promote damage repair, while avoiding the risks of safety brought about by cell transplantation, which is a safe and effective potential treatment. This review comprehensively discusses ADMSC-related therapies in CNS diseases, highlighting their advancements and outlining their promising prospects as emerging therapies.

## 2. Search Strategy and Selection Criteria

We conducted a comprehensive search in the PubMed database using the keywords “adipose-derived mesenchymal stromal cells” or “adipose-derived mesenchymal stem cells” or “genetic modified ADMSC” in conjunction with other specific terms/diseases, including stroke, traumatic brain injury, spinal cord injury, epilepsy, Alzheimer’s disease, Parkinson’s disease, amyotrophic lateral sclerosis, multiple-system atrophy, multiple sclerosis, exosome, secretome, conditioned medium, and clinical trial. Our search strategy aimed to identify relevant articles published from inception to October 2023. We excluded articles that did not pertain to ADMSC or the therapy of CNS disorders listed. Following the database search, duplicate entries were removed, and the remaining articles were screened by title and abstract for relevance. Full-text articles were then assessed for eligibility based on predefined inclusion and exclusion criteria, which revolved around the application of adipose-derived mesenchymal stromal/stem cells in the treatment of central nervous system disorders.

## 3. Mechanism of ADMSC-Related Therapy in Central Nervous System

ADMSC-related therapy encompasses ADMSC transplantation and treatment with ADMSC-derived exosomes. Given that neurological diseases frequently entail dysfunction or loss of neural cells [26], a promising approach to address this issue involves the transdifferentiation of ADMSCs to obtain neural stem cells (NSCs) or neural progenitor cells (NPCs), followed by their transplantation into the affected regions. Notably, the efficacy of ADMSC transplantation is contingent upon the condition of both the donor and recipient [27]. Factors such as body mass index, diabetes, advanced age, exposure to radiotherapy, and Tamoxifen can adversely affect the proliferation and differentiation capacity of ADMSCs. Conversely, higher circulating estrogen levels have demonstrated a positive correlation with improved graft viability and enhanced adipocyte function [27].

In order to further uncover the mechanism of ADMSC transplantation, extensive research has identified essential transcription factors, chemicals, growth factors, and regulatory RNAs involved in the transdifferentiation of ADMSCs into neural cells [28]. These factors collectively modulate crucial signaling pathways that regulate gene transcription and expression. For instance, it has been reported that Wnt5a, a noncanonical Wnt pathway, promotes neural differentiation of ADMSC by mediating the activation of JNK and up-regulating the expression of target genes Fzd3 and Fzd5 [29]. The Notch pathway is pivotal for hADMSCs as it suppresses adipocyte differentiation and maintains self-renewal [30]. Downregulation of the Notch pathway promotes transdifferentiation, primarily influenced by the inducing microenvironment [31]. Moreover, the Notch pathway also plays a role in NSC proliferation and self-renewal in both in vitro and in vivo settings [32]. Activation of Notch receptors enhances NSC survival by triggering cytoplasmic signaling, including STAT3 and Akt, resulting in upregulation of sonic hedgehog (Shh) and hairy and enhancer of split 3 (HES3) levels [33]. The Shh pathway, which is active in hADMSCs, also contributes significantly, with its inhibition positively impacting the differentiation of hADMSCs [34,35] (Figure 1).

## 4. ADMSCs and Therapy of CNS Injuries

### 4.1. ADMSCs-Related Therapy in Stroke

Stroke, a prevalent central nervous system (CNS) injury, ranks as the second leading cause of death and a major contributor to disability [36,37]. Considering the global health burden imposed by stroke, especially in developing countries, the imperative for more effective treatments to enhance post-stroke management becomes increasingly urgent [38]. ADMSCs present a promising and innovative therapeutic approach for stroke intervention, facilitating improved post-stroke neuronal survival and cerebral function primarily through immunomodulation and the inhibition of neuronal apoptosis. In a study by Huang et al., intra-arterial transplantation of ADMSCs into rats with middle cerebral artery occlusion (MCAO) facilitated the regeneration of blood vessels and neuronal fibers in the peri-infarct cortex, resulting in improved neurological function [39]. The observed clinical benefits were, in part, ascribed to the upregulation of key regulators, specifically netrin-1 and its receptor, which played a significant role in the remodeling of both vascular and neuronal networks within the peri-infarct cortex [39]. Notably, Zhou et al. demonstrated that transplantation of human ADMSCs (hADMSCs) in MCAO mice promotes memory, spatial learning, and stroke symptom amelioration by transdifferentiating into neuron-like cells (MAP2+) in vivo, thereby establishing hADMSCs as a potential replacement-based stem cell therapy for stroke [40]. Moreover, hADMSCs also acted as immunomodulators to enhance immunosuppression and improve the survival of endogenous neurons [40]. In order to make ADMSC reach the lesion accurately and improve treatment efficiency, Zhang et al. used stereotactic transplantation of ADMSCs on the mouse model of intracerebral hemorrhage (ICH) and found that it led to significant functional improvement, as well as alleviated brain edema and suppressed cell apoptosis [41]. This mechanism was linked to the upregulation of AQP4 protein expression, which modulated inflammation and brain edema through mitogen-activated protein kinase (MAPK) pathways, including the c-Jun N-terminal kinase (JNK) pathway and p38/MAPK pathway [41].

Nonetheless, ADMSC transplantation inherently carries a tumorigenic risk, and the relatively low survival rate of the transplanted cells introduces uncertainty in its efficacy. On the other hand, the administration of cell-free extracts from ADMSCs (ADMSC-E) exhibited faster clinical improvement and a more robust immunomodulatory effect. Despite the similar long-term clinical benefits observed with both approaches [42], the utilization of ADMSC-E appears to be a safer alternative, holding significant promise for the clinical treatment of ischemic injuries. Zhao et al. observed that transient MCAO rats treated with intravenous injection of ADMSC-E exhibited significantly reduced ischemic lesions and fewer apoptotic neurons compared to other control groups. The therapeutic effect was primarily attributed to the protein component of ADMSC-E [42]. The therapeutic efficacy of ADMSC-E can be attributed, at least in part, to the presence of micro-RNAs within its composition. Studies have shown that ADMSC-E enriched with microRNA (miR)-30d-5p can inhibit the inflammatory response by reversing microglial polarization induced by oxygen–glucose deprivation (OGD) through autophagy modulation in vitro. In vivo, these exosomes promote post-MCAO microglial polarization to M2, suppressing autophagy-mediated inflammatory responses and alleviating MCAO-induced brain injury [43]. Furthermore, miR-31 secreted by ADMSCs-derived extracellular vesicles downregulated the translation of TRAF6, subsequently decreased expression of interferon regulatory factor 5 (IRF5), and mitigated neuronal damage, thereby inhibited neuronal apoptosis in MCAO mice models, reduced post-ischemic infarct volume, improved neuronal cell survival after OGD, and finally promoted neurological function recovery after ischemic stroke [44]. The latest research showed that uptake of intranasal administrated ADMSC-E by neuron could inhibit post-ischemic neural ferroptosis in MCAO mice and improve neurobehavior function by downregulation of ChaC glutathione-specific γ-glutamylcyclotransferase 1 (CHAC1), which was reported as a downstream target gene of miR-760-3p in N2a cells [45] (Figure 2). While studies on the potential application of ADMSCs in stroke treatment show promise, there remains a substantial journey ahead before clinical implementation can be realized.

### 4.2. ADMSCs-Related Therapy in Traumatic Brain Injury

Traumatic brain injury (TBI) is a prevalent form of CNS injury, and the current treatment strategies primarily involve supportive care, controlling intracranial hypertension, and maintaining cerebral perfusion pressure [46]. The application of ADMSC-E emerges as a promising strategy to mitigate injury and foster functional recovery. Studies have demonstrated that the intranasal delivery of hADMSC-E 48 h after traumatic brain injury (TBI) or intravenous administration at 3 h post-injury effectively attenuated cognitive and motor impairments [47]. This could be attributed to the regulatory molecules present in exosomes, effectively modulating inflammatory cell activation and response, thereby reducing cell death and cortical brain injury. Studies have shown that intravenous infusion of ADMSCs’ secretome in rats with electric cortical contusion impactor-induced TBI leads to enhanced neurological functional recovery, reduction of brain tissue vasogenic edema and nerve fiber damage, suppression of TBI-induced inflammation, and neural cell apoptosis. This is accompanied by the downregulation of tumor necrosis factor-alpha (TNF-α) and interleukin-6 (IL-6), while tumor necrosis factor-stimulated gene 6 protein (TSG-6) and transforming growth factor-beta (TGF-β) are upregulated [48]. In a TBI rat model induced by weight-drop, treatment with intracerebroventricular microinjection of hADMSC-derived exosomes (hADMSC-E) results in their uptake primarily by microglia/macrophages, leading to the inhibition of their activation and playing a protective role. This treatment approach improves functional recovery, reduces neuronal apoptosis, suppresses neuroinflammation, and promotes neurogenesis. The therapeutic effects of hADMSC-E are comparable to those of hADMSCs themselves [49]. It is noteworthy that the therapeutic effect of hADMSC-E may be influenced by the age of the treated individuals. Intravenous transplants of hADMSC grafts have been shown to ameliorate TBI-associated motor asymmetry, forelimb akinesia, hippocampal cell loss, and impact and peri-impact area in young TBI rats, whereas these effects are reduced or absent in aged animals. Moreover, the improvement of TBI-induced cognitive impairments is diminished in aged animals compared to young animals, potentially attributed to the decreased homing of cells to the spleen in aged individuals [50].

Long non-coding RNAs (lncRNAs), particularly metastasis-associated lung adenocarcinoma transcript 1 (MALAT1), play a crucial role in the beneficial effects of hADMSC-E [47,50,51]. Intravenous administration of hADMSC-E containing MALAT1 in TBI rats leads to significant motor behavior recovery, reduction of cortical brain injury, and modulation of multiple pathways involved in inflammation, cell death, cell cycle regulation, expression of other noncoding RNAs, and regenerative molecular pathways. The therapeutic effects are not observed when conditioned media depleted of exosomes or exosomes depleted of MALAT1 are used, emphasizing the pivotal role of MALAT1 in the regenerative effects mediated by hADMSC exosomes [51]. NRTK3 (TrkC) has been identified as one of the regulatory targets of MALAT1 in hADMSC-E, where MALAT1 modulates the expression of full-length TrkC, activating the MAPK pathway and facilitating recovery [47]. Knockdown of lncRNAs MALAT1 and nuclear-enriched abundant transcript 1 (NEAT1) impairs the motor behavior amelioration mediated by conditioned media and cognitive improvements mediated by both hADMSCs and conditioned media in young TBI rats [50].

### 4.3. ADMSCs-Related Therapy in Spinal Cord Injury

Spinal cord injury (SCI) occurs as a result of neuronal axon damage caused by various factors, leading to impaired neural transmission and dysfunction in sensation and voluntary movements below the affected area [52]. Nevertheless, the damage and impaired functionality of neurons, compounded by their inherent challenge in regeneration, present a substantial obstacle in the treatment of spinal cord injuries. In this context, ADMSC-related therapy offers a promising approach to mitigate neuronal damage in spinal cord injuries and fosters the generation of new neurons, thereby representing a potential therapeutic strategy to promote nerve regeneration and repair. In the context of spinal cord injury (SCI), ADMSC transplantation has been shown to upregulate the expression of procollagen-lysine, 2-oxoglutarate 5-dioxygenase 2 (PLOD2), which is a lysyl hydroxylase enzyme [53,54]. Notably, additional investigations conducted by Li et al. have corroborated that the neuroprotective effect elicited by ADMSCs in vitro is dependent on PLOD2 activity and occurs through the mediation of the TGF-β1/Smad3/PLOD2 pathway, as observed in a rat SCI model [55]. Collagen, being the regulatory target of PLOD2, assumes a pivotal role in various cellular processes, such as cell–extracellular matrix (ECM) interactions, cell signaling, adhesion, as well as the availability and stability of growth factors crucial for the survival, differentiation, and axonal outgrowth of neurons [56]. Hence, the up-regulation of PLOD2 expression can have a significant impact on the stability of intermolecular collagen cross-links, subsequently influencing the microenvironment’s stability at the lesion site. Another mechanism through which ADMSCs exert their neuroprotective role in SCI involves the inhibition of neuroinflammation [57]. The improved neuronal function and survival are associated with reduced expression of proinflammatory regulators and inhibition of the Jagged1/Notch signaling pathway [57]. Transplantation of ADMSCs through lumbar puncture in dogs with paraplegia due to SCI resulted in clinical recovery, as evidenced by improved voluntary contraction during hind limb loading as observed through percutaneous electromyography [58]. ADMSC infusion in a blunt SCI rat model showed that injection of ADMSC distal to the injured area was an administration with satisfactory safety and effect, which could significantly decrease neuronal death, but could not inhibit the loss of myelin or control the formation of a scar as it could not reduce astrocyte’s migration towards the injury area [59]. Furthermore, autologous transplantation of ADMSCs has been attempted in patients with acute SCI, demonstrating effectiveness and safety, with significant improvements in neurological function and quality of life. However, the rate of recovery remains modest [60]. Although ongoing research is advancing our understanding of the mechanism and application of ADMSC-related therapy for SCI, further studies are needed to enhance the efficacy of this treatment approach.

## 5. ADMSCs-Related Therapy of Epilepsy

Epilepsy is a chronic brain disease characterized by persistent seizures and abnormal behaviors and sensations [61]. The reported incidence rate of epilepsy is 61.44 per 100,000 person-years (95% CI 50.75–74.38), with an annual cumulative incidence of 67.77 per 100,000 persons (95% CI 56.69–81.03) [62]. Notably, two incidence peaks are observed: one in the pediatric population aged 0 to 5 years [63], and another in senior citizens aged >75 years (up to 9.7/1000) [64]. Although medication is the most common treatment for epilepsy [65], recurrent seizures can still occur despite appropriate medication. Both preclinical and clinical research have suggested the preliminary efficacy and safety of MSC therapy for drug-resistant epilepsy, necessitating further studies to explore its therapeutic potential.

In a previous study, reduced levels of B-cell lymphoma 2 (Bcl-2) protein were detected in the brains of KA-induced epileptic rats, particularly within 24–48 h after the onset of seizures [66]. To investigate this further, Wang et al. conducted a transplantation of differentiated ADMSCs into the hippocampus of KA-induced rats. The results revealed that ADMSC transplantation could effectively reduce KA-induced seizures and promote the recovery of memory and learning capacity in rats with epilepsy. The neuroprotective function of ADMSCs in epileptic rats was correlated with the upregulation of neuroprotective cytokines and apoptotic inhibitors, such as B-cell lymphoma-extra-large (Bcl-xL) and Bcl-2, while simultaneously downregulating the expression of the pro-apoptotic protein Bcl-2-related X protein (Bax) [67]. This study provided valuable insights, suggesting that the upregulation of Bcl-2 expression and the downregulation of Bax could be crucial mechanisms underlying the neuroprotective effect of ADMSCs in epilepsy. Bax, known as a promoter of apoptosis, is overexpressed in the brains of epileptic rats [68]. Bcl-xL plays a protective role in epilepsy by interacting with pro-apoptotic proteins such as Bax and Bad. However, in the post-seizure hippocampus, increased neural clusterin binds to Bcl-xL, which correlates with neuronal apoptosis and hippocampal injury induced by prolonged seizures [68]. Further study is needed to elucidate the role of increased Bcl-xL induced by ADMSC transplantation.

Gao et al. have identified that increased circHivep2 is a protective factor that reduces the expression of inflammatory factors and the activation of microglial cells in vitro. In vivo application of circHivep2+ exosomes derived from ADMSCs prior to inducing epilepsy through KA injection resulted in decreased behavioral seizure scores. The biological regulatory function of circHivep2 is achieved through the miR-181a-5p/SOCS2 pathway. Overexpressed circHivep2 downregulates the level of miR-181a-5p, thereby enhancing the expression of SOCS2 while decreasing the levels of inflammatory factors such as IL-1β and TNF-α [69].

## 6. ADMSCs-Related Therapy of Neurodegenerative Diseases

Neurodegenerative disorders, including Alzheimer’s disease and Parkinson’s disease, are characterized by neuronal damage leading to cognitive, memory, and mobility dysfunction. Currently, therapeutic options for these diseases are limited. The pathogenesis of neurodegenerative disorders involves gradual chronic neuronal cell death and impaired neuronal transmission or myelin sheath deterioration [70,71]. For these diseases, there are still no available drugs or therapy could promise an entire recovery [72]. ADMSCs-related therapy holds significant promise in this field, and several studies and clinical trials are currently underway. This section provides a review of adipose-derived stem cell therapy in neurodegenerative disorders, focusing on Parkinson’s disease (PD) and Alzheimer’s disease (AD) as examples.

Autologous ADMSCs are regarded as the most suitable material for PD-related therapy, given that their application addresses ethical concerns [73]. ADMSC treatment has been found to exert long-term effects attributed to neuroprotection and neurogenesis mechanisms [74]. To evaluate therapeutic effects and side effects in PD studies, PD animal models induced by neurotoxins, particularly the rodent 6-hydroxydopamine (6-OHDA) model, are commonly employed as preclinical assessment tools [75]. In a meta-analysis conducted by Li et al., it was demonstrated that neural-induced ADMSCs exhibited more favorable clinical outcomes in animal models compared to primitive ADMSCs, while other researchers reported that primitive ADMSCs showed greater effectiveness in neuroprotection [74]. ADMSC transplantation results in increased levels of neurotrophic factors and alterations in the microenvironment, which contribute to the establishment of an in vivo nutritional niche and supplementary mechanism [76]. Additionally, ADMSC-derived exosomes show promise as vectors in this therapy, both in vitro and in vivo. Recent research has elucidated that the utilization of miR-188-3p-enriched exosomes derived from ADMSCs exhibits inhibitory effects on pyroptosis and autophagy, concurrently stimulating proliferation. The specific targets of miR-188-3p have been identified as NOD-like receptors 3 (NLRP3) and cyclin-dependent kinase 5 (Cdk5), signifying its promising potential as a therapeutic target in Parkinson’s disease (PD) [77]. Moreover, secreted molecules released by neural-induced human adipose tissue-derived stem cells (NI-ADMSC-SM) could reduce the dysfunction of endoplasmic reticulum and chondriosome induced by rotenone, which was similar to the cytopathy in PD, indicating that NI-ADMSC-SM has neuroprotective effects and the ability to improve neurofunction of PD patients [78].

Drug delivery techniques represent a prominent research focus in the application of ADMSC-related therapy for Parkinson’s disease (PD). At present, intracerebral injection is considered the most viable administration route, as it minimizes ADMSC damage and maximizes the potential for recovery [74]. Concurrently, ongoing research on technologies associated with this treatment is continuously advancing. The Super Paramagnetic Iron Oxide Nanoparticle (SPION)/poly-L-lysine hydrobromide (PLL) transfection method has been identified as suitable in ADMSC-related therapy [79]. Researchers have also suggested that pre-transplantation stimulation of ADMSCs with n-Butylidenephthalide (BP) extracted from Angelica sinensis in the striatum enhances the motor function recovery of 1-methyl-4-phenyl-1,2,3,6-tetrahydropyridine (MPTP)-induced PD mice. Gene expression analysis revealed the overexpression of genes associated with neurogenesis after incubation with BP [80].

The accumulation of β-amyloid peptide (Aβ) in brain tissue represents a critical pathological feature of Alzheimer’s disease (AD). ADMSC-related therapy has demonstrated its efficacy in effectively reducing Aβ accumulation, promoting the activation of neural cells, and mitigating inflammatory responses. Studies have reported that transplantation of adipose-derived stem cells (ADMSCs) in APP/PS1 double transgenic AD model mice significantly reduces Aβ deposition, improves cognition, and enhances memory and learning capabilities. Additionally, transplantation leads to enhanced activation of microglial cells in the cortex and hippocampus, along with alleviation of inflammation. This is accompanied by upregulation of alternative activation markers and enzymes involved in Aβ degradation, highlighting the potential of ADMSC transplantation as a therapeutic approach for AD [81]. Furthermore, ADMSC-derived exosomes enriched with Neprilysin (the primary enzyme responsible for Aβ degradation in the brain) demonstrate the ability to reduce intracellular and secreted Aβ levels when transferred into N2a cells. Notably, the therapeutic effect of exosomes derived from ADMSCs surpasses that of exosomes from bone marrow-derived mesenchymal stem cells [16].

Amyotrophic lateral sclerosis (ALS) is a progressive, neurodegenerative disorder chiefly impacting the motor neurons of the spinal cord, brainstem, and brain, culminating in paralysis. In the absence of effective treatment modalities, the median survival duration for individuals afflicted with ALS ranges between 2 to 4 years [82]. The scientific community has been actively exploring potential breakthroughs in ALS treatment emanating from ADMSCs. A hallmark clinical characteristic of familial ALS (FALS) is the mutation in Superoxide Dismutase 1 (SOD1) [83]. Lee et al. investigated the effect of ADMSC-E on the neuronal cells of G93A ALS mice in vitro, finding that ADMSC-E significantly reduced both the production and accumulation of intracellular SOD1, improving the mitochondrial dysfunction present in neuronal cells [17]. In a singular case report, a male patient, who manifested ALS symptoms at the age of 46, was administered a total of six autologous ADMSC intravenous infusions over a span of 4 to 7 years post-symptom onset and exhibited sustained well-being 7 years post-initial administration. This suggests the potential efficacy of ADMSC transplantation in either enhancing clinical status or decelerating disease progression in certain ALS patients [84]. On the other hand, a single-center, prospective, open-label, single-arm clinical trial involving allogeneic ADMSC transplantation in 17 ALS patients showed that, while no adverse events related to ADMSC transplantation were found, confirming the feasibility and safety of this treatment, there was no evidence to suggest that the transplantation could reverse or slow down the progression of ALS [85].

Multiple-system atrophy (MSA) is a rare, progressive neurodegenerative condition in adults, manifesting with ataxia, parkinsonism, and autonomic disturbances. A study by Chang et al. delved into the effects of intracerebral ADMSC injection in transgenic mice. It was reported that the transplantation of ADMSCs (at a dosage of 2 × 10^5^ ADMSCs per mouse) notably improved the rotarod performance of the subjects [86]. Subsequent in vitro examinations revealed that intracerebral transplantation of ADMSCs could mitigate striatal degeneration and inflammation, improve the nigrostriatal pathway for dopamine transmission, and enhance cell survival as well as myelination within the caudate-putamen region [86] (Table 1).

## 7. ADMSCs-Related Therapy of Multiple Sclerosis

Multiple sclerosis (MS) is one of the most common demyelinating diseases and among the most devastating conditions affecting the central nervous system [87]. ADMSC-related therapy holds promise as a safe and effective approach to treat MS, as no serious adverse effects have been reported in either clinical or preclinical research studies investigating ADMSCs-related treatments [88,89,90,91]. The pathological characteristics of MS, mainly reflecting an autoimmune response, can be replicated in the murine experimental autoimmune encephalomyelitis (EAE) model [92]. Hedayatpour et al. suggested that intravenous transplantation of ADMSCs into the demyelinated corpus callosum in the mouse EAE model could promote remyelination, enhance motor function, and reduce inflammation [89]. Significantly, clinical trials have provided evidence of the safety of intrathecal ADMSC treatment for patients with multiple sclerosis (MS). However, it is essential to exercise caution and reserve its application for severe cases, aggressive disease progression, and instances in the inflammatory phase [88]. Considerations regarding donor demographics are crucial when selecting suitable ADMSCs for treatment, as obesity has been found to limit or compromise the anti-inflammatory effects of human ADMSCs [91]. Hence, ADMSCs derived from obese donors are not an appropriate source for the treatment of autoimmune diseases like MS.

Moreover, the microenvironment surrounding ADMSCs contains immunomodulators and neuroprotective factors, which also contribute to the pathological improvement of MS. Yousefi et al. demonstrated that ADMSC-conditioned medium, in combination with ADMSCs, functions as an immunomodulator and neuroprotector in a mouse experimental autoimmune encephalomyelitis (EAE) model [93]. Additionally, the stromal vascular fraction (SVF) of adipose tissue, consisting of ADMSCs, leukocytes, and adipocytes [94,95,96], exhibits significant potential for MS treatment due to its effective regulation of inflammatory factors. Importantly, the amelioration of CNS pathology induced by SVF is more pronounced than that achieved by ADMSCs alone [97]. These studies underscore the significant potential of regulatory factors present in the microenvironment surrounding ADMSCs in the context of MS therapy.

## 8. The Application of Gene-Modified ADMSC in CNS Disorders

With the continuous advancement in gene modification technology and the deepened understanding of ADMSCs, researchers have begun employing gene modification to explore further therapeutic possibilities associated with ADMSCs. For instance, Nan et al. utilized lentivirus to transduce brain-derived neurotrophic factor (BDNF) into ADMSCs in vitro, thereby establishing the feasibility of using ADMSCs as a delivery vector for BDNF. This work lays a foundational stone for leveraging ADMSCs in the treatment of CNS disorders [98]. The lentivirus-transfected ADMSCs, which overexpressed BDNF and neurotrophin-3 (NT-3), were found to promote the differentiation of ADMSC into neurons, with a noted synergistic effect between BDNF and NT-3 [99]. An in vivo study conducted on dogs demonstrated that post-spinal cord injury (SCI) transplantation of ADMSCs overexpressing BDNF (BDNF-ADMSC) and ADMSCs overexpressing heme oxygenase-1 (HO1-ADMSC) significantly contributed to functional recovery, inflammation reduction, and restoration. Specifically, HO1-ADMSC mitigated inflammation, while BDNF-ADMSC spurred neuroregeneration, with a synergistic effect observed between the two [100]. In another study, rat ADMSCs overexpressing neurogenin2 (Ngn2-ADMSCs) were transplanted into rats with SCI. The transplanted cells were found to localize around the lesion and differentiate into neurons, which facilitated post-SCI functional recovery. This approach was found to be superior to transplantation with non-transfected ADMSCs in improving the local microenvironment post-injury [101].

Given the pivotal role of neuroinflammation in the onset of CNS disorders, an increasing number of studies have shifted focus towards employing ADMSCs as vectors for anti-inflammatory agents. This approach facilitates sustained expression of anti-inflammatory cytokines, presenting a potential therapeutic strategy for neuroimmune disorders such as multiple sclerosis. Researchers have demonstrated that the transduction of IFN-β into ADMSCs enables the consistent secretion of interferon (IFN)-β without compromising the differentiation and secretion capabilities of the ADMSCs. As a gene therapy modality, IFN-β-ADMSCs exhibited superior anti-inflammatory effects compared to non-transfected ADMSCs in an experimental autoimmune encephalomyelitis (EAE) mouse model [102]. A study conducted by Mohammadzadeh et al. on EAE mice revealed that IFN-β-MSC significantly diminished interleukin (IL)-17 levels, induced IL-10 and Treg, and mitigated cell infiltration in brain tissue, further corroborating the anti-inflammatory potential of IFN-β-ADMSCs [103]. IL-4, another crucial anti-inflammatory cytokine, when overexpressed in ADMSCs (IL-4-ADMSCs) and transplanted early into EAE mice, significantly lowered the clinical score of EAE compared to controls and non-modified ADMSC groups, without altering the phenotype and functional attributes of ADMSCs. This suggests a potential avenue for controlling neuroinflammation and alleviating clinical symptoms [104].

Transgenic ADMSCs, capable of differentiating into neurons and secreting neuroprotective cytokines, emerge as a promising strategy for treating neurodegenerative diseases. It has been reported that glial cell-derived neurotrophic factor (GDNF)-overexpressing ADMSCs (GDNF-ADMSCs), engineered via lentivirus transfection, could morph into neuron-like cells in vitro. Post-transplantation, GDNF-ADMSCs were viable for at least 90 days and differentiated into dopaminergic neuron-like cells, significantly ameliorating the clinical symptoms in 6-OHDA-induced PD rats [105]. Utilizing a non-viral Sleeping Beauty (SB) transposon system, Stahn et al. engineered GDNF-ADMSCs and introduced them into a mild 6-OHDA hemiparkinson male rat model. The findings highlighted the immunomodulatory and neuroprotective attributes of GDNF-ADMSCs, including the restoration of tyrosine hydroxylase-expressing cells [106] (Table 2).

## 9. The Application of ADMSCs Secretome in CNS Disorders

The secretome of ADMSCs, a complex mixture secreted by ADMSCs into the surrounding microenvironment, is attracting attention as a promising cell-free therapeutic avenue, with a growing body of research exploring its therapeutic potential in CNS disorders [14]. In models of CNS disorders, the ADMSC secretome primarily serves as a neuroprotective agent, promoting repair, reducing inflammation, and decreasing cell death. A study reported the neuroprotective efficacy of ADMSC secretome on SH-SY5Y cells subjected to arsenic in vitro, where it decreased cell death and fostered neural functional restoration, albeit without promoting the differentiation of SH-SY5Y cells into a mature neuron-like phenotype [107]. Another investigation revealed that the ADMSC secretome could significantly attenuate lipopolysaccharide (LPS)-induced microglial activation and decrease microglia-mediated neuroinflammation in vitro, implicating the involvement of sphingosine kinase/S1P signaling in this regulation [107]. Baldassarro et al. elucidated that the composition, biological attributes, and therapeutic potential of ADMSC secretome are, to some extent, donor-dependent [108]. Particularly, an ADMSC secretome enriched with BDNF notably reduced the death of OGD-induced neurons, augmented the differentiation of mature oligodendrocytes, and facilitated neural restoration [108]. Considering the potential alterations in secretome composition in response to microenvironmental changes during certain disease states, researchers have commenced investigations into the impact of pathological microenvironments on ADMSC secretome. Upon treatment with post-traumatic brain injury cerebrospinal fluid (post-TBI CSF), rendering them TBI-ADMSCs, a rise in mRNA levels of both pro-inflammatory and anti-inflammatory genes was observed. When THP1 macrophages were exposed to the secretome produced by TBI-ADMSCs, a further induction to differentiate and mature was noted, evidenced by an increased proportion of CD11b+, CD14+, and CD86+ cells alongside elevated phagocytosis activity of CD14+ and CD86+ cells [109]. However, a downregulation of both pro- and anti-inflammatory genes in macrophages was reported [109]. These findings suggest that the ADMSC secretome may undergo complex alterations in pathological microenvironments, exerting diverse effects on the microenvironment concurrently.

## 10. The Prospect of ADMSC Related Therapy and Clinical Trails

At present, there are not only increasing studies on ADMSCs-related therapy in the treatment of CNS diseases, but also some clinical trials. The clinical trials primarily concentrate on evaluating the safety of ADMSC-related therapies, their efficacy in CNS diseases, and advancements in optimal dosing and delivery strategies. For example, a clinical trial demonstrated the efficacy and safety of autologous ADMSCs (GXNPC-1) transplantation in patients with chronic stroke. The results indicated notable improvement in the condition of all participants, and no safety concerns were observed during the 6-month period following implantation [110]. The phase II study, which was an open-label, single-center, sequential study that explored the appropriate dose of GXNPC-1 transplantation was recently completed (NCT numbers: NCT04088149). A study with 47 SCI patients recruited indicated that autologous adipose-derived MSCs transplantation was safe and was efficient for the post-SCI improvement of neurological muscle and neurogenic bladder [60]. Moreover, researchers are currently investigating the potential of autologous ADMSCs to treat patients with mild traumatic brain injury (TBI), specifically post-concussion syndrome, aiming to mitigate severe outcomes such as Alzheimer’s disease (AD) and chronic traumatic encephalopathy(CTE) [111] (NCT numbers: NCT04744051). Furthermore, researchers have investigated the impact of different dosages and appropriate doses of ADMSC transplantation for stroke therapy (NCT numbers: NCT03570450 and NCT04088149).

While significant progress has been made in exploring ADMSC-related therapy for CNS diseases, the safety and effectiveness of clinical ADMSC-related therapy are yet to be fully elucidated. In order to bridge the gap between research and clinical application, enhancements in the administration methods and precise targeting of ADMSC-related therapy are necessary. Furthermore, additional efforts are required to optimize efficacy and control potential side effects. Many chronic CNS diseases, including neurodegenerative disorders, necessitate multiple doses over an extended period, and invasive administration of ADMSCs carries the risk of surgical injury [72]. Therefore, it is crucial to develop less invasive methods of administration that are suitable for long-term application. Additionally, the degenerating tissues may lack a specific homing signal for transplanted ADMSCs, which can potentially diminish the therapeutic effect in neurodegenerative diseases [112]. It has been found that removing chondroitin sulfate from the extracellular matrix enhances the interaction between stem cells and host tissue, offering a potential improvement for the treatment of currently incurable neurodegenerative diseases [113]. Moreover, while exosomes derived from ADMSCs can transport multiple molecules into tissues, achieving accurate targeting remains a challenge. Regardless of the carried molecules being pharmacological or regulatory, it is essential to maximize the precision of the therapeutic effect in order to enhance efficacy and reduce side effects. Although previous studies have highlighted the potential of ADMSC-related therapies as a new approach for treating CNS diseases, further rigorous in-depth studies and discussions are warranted to advance our understanding (Table 3).

## 11. Conclusions

Significant advancements have been made in exploring ADMSC-related therapy for CNS disorders. This therapeutic approach encompasses ADMSC transplantation and the utilization of ADMSC-derived exosomes and secretome, with key mechanisms including immune regulation, inhibition of inflammatory responses, reduction of neuronal injury and apoptosis, and promotion of neural regeneration. ADMSC-related therapy holds significant promise as a potential method to ameliorate damage and enhance neurological function in central nervous system (CNS) disorders, benefiting from the regenerative and restorative capabilities of stem cells, coupled with their favorable safety profile and easy accessibility, as well as the ability of microenvironmental modulation by paracrine. However, despite numerous basic studies demonstrating the feasibility of ADMSC-related therapies in CNS disorders, further research is essential to establish the safety and effectiveness of clinical ADMSC-related therapy. Moreover, aiming for an improved clinical approach, enhancements in administration methods and exploration for appropriate dose are necessary to minimize invasive damage and enhance therapeutic precision. These aspects warrant continuous investigation and attention to propel the field of ADMSC-related therapy for CNS disorders forward.

## Figures and Tables

**Figure 1 pharmaceutics-15-02637-f001:**
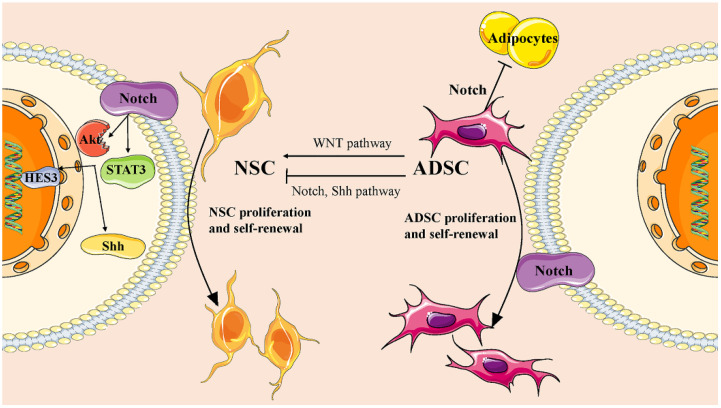
The mechanism of ADMSC-related therapy in central nervous system.

**Figure 2 pharmaceutics-15-02637-f002:**
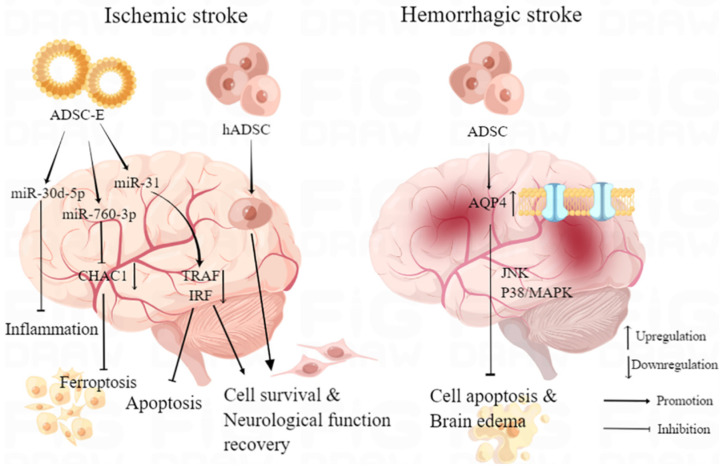
ADMSCs-related therapy in stroke and mechanism. By Figdraw.

**Table 1 pharmaceutics-15-02637-t001:** Summary of experimental studies investigating the application of adipose-derived MSCs in neurodegenerative diseases.

Neurodegenerative Diseases	Animals or Cells Model of Disease	Origin of Graft	Type of Therapy	Summarized Experimental Design	Main Outcomes/Therapeutic Effect	Reference
Parkinson’s disease	6-OHDA rat model	Autologous ADMSC	ADMSC transplantation	Investigating ability of both naive and differentiated ADMSC autologous graft to protect, repair or restore the function of nigrostriatal pathway was performed on the basis of study of neurohistology, electrophysiology, cell biology, and gene expression.	Produce neuroprotective trophic factors, modulate the microenvironment	[76]
Parkinson’s disease	MPTP-induced mice models; MPP+-induced cell models	Adipose obtained from C57BL/6 mice	Injection of miR-188-3p-enriched exosome derived from ADMSC	The level of miR-188-3p was assessed in PD patients. Levels of injury, inflammatory factors and autophagy were evaluated in MPP+-induced cell models and MPTP-induced PD mice models after treated with miR-188-3p-enriched ADMSC-E.	Inhibit autophagy, pyroptosis and promote proliferation by targeting CDK5 and NLRP3 in vitro. Alleviate inflammation and restore neurons in the substantia nigra in vivo.	[77]
Parkinson’s disease	ROT-induced SH-SY5Y cells	Adipose obtained from human donors	Injection of NI-ADMSC-SM	Neuroprotective effects of NI-ADMSC-SM on ROT-induced dysfunction in human SH-SY5Y cells was explored by measuring the alternation of damage in endoplasmic reticulum, the mitochondria, and their tethering proteins with molecular biological method.	Restore mitochondrial fusion, mitophagy, and tethering to ER	[78]
Parkinson’s disease	6-OHDA rat model	Adipose obtained from male Sprague–Dawley rats	ADMSC transplantation	Adipose-derived MSCs obtained were immunostained-coating SPION with polyl-lysine hydrobromide and transfecting GFP reporter gene into ADMSCs to explore the role of external magnets in the delivery and homing of stem cells in the target tissue.	Provide a solution for the delivery and homing of transplanted adipose-derived MSC transplantation	[79]
Parkinson’s disease	MPTP-induced mice models	Adipose obtained from human donors	ADMSC transplantation	Evaluation of the possible neurogenic effects of BP was performed in vitro, including examination of cell survival and gene expression patterns. After transplanting ADMSCs into the mouse striatum, effects of BP-pretreated ADMSCs in vivo were evaluated by behavioral experiment.	Restore motor abilities. BP stimulation improved the therapeutic effects of transplantation.	[80]
Alzheimer’s Disease	APP/PS1 double transgenic AD model mice	Adipose obtained from male Sprague–Dawley rats	ADMSC transplantation	Immunomodulatory effect and mechanism of microglia activation after adipose-derived MSC transplantation was revealed in AD model mice. The effects of ADMSC transplantation on cognitive function of AD mice were studied by behavioral experiments.	Optimize alternative microglial activation, alleviate inflammation and Aβ pathology. Improves cognition, memory, and learning capabilities	[81]
Amyotrophic lateral sclerosis	G93A ALS mice model	Subcutaneous adipose obtained from human donors	Neuronal cells of ALS mice were treated with ADMSC-E in vitro	The production and aggregation of SOD1, as well as the mitochondrial function was evaluated after treatment with ADMSC-E for twice in vitro.	Inhibit production and aggregation of intracellular SOD1, alleviate the mitochondrial dysfunction of neuronal cells.	[17]
Multiple-system atrophy	MBP Line 1 (MBP1) mice	Human ADMSCs	Intracerebral injection of ADMSCs	The effective dose of ADMSC transplantation was explored, and immunohistochemical and molecular biological methods were used to investigate the mechanism.	Alleviate striatal degeneration and inflammation, improve the nigrostriatal pathway for dopamine, ameliorate cell survival and myelination at caudate-putamen.	[86]

ADMSC: adipose-derived mesenchymal stem cell; 6-OHDA: 6-hydroxydopamine; MPTP: 1-methyl-4-phenyl-1,2,3,6-tetrahydropyridine; MPP: methyl-4-phenylpyridinium; ROT: Rotenone; NI-ADMSC-SM: Neural-induced ADMSC secretome; SPION: Super Paramagnetic Iron Oxide Nanoparticle; BP: n-Butylidenephthalide; APP: amyloid precursor protein; PS1: presenilin 1; SOD1: Superoxide dismutase 1; MBP: Myelin-based protein.

**Table 2 pharmaceutics-15-02637-t002:** Exploration of the therapeutic potential of gene-modified ADMSCs in CNS disorders.

CNS Disorders	Gene-Modified ADMSCs	Animal Model of Disease	Origin of Transplanted ADMSCs	Method of Administration	Main Therapeutic Effect	Reference
Spinal cord injury	BDNF-ADMSC	Male SCI dog model	Gluteal subcutaneous fat from healthy beagle dog of age 1.5 years	Two injections into epicenter and one into the center of the injured segment	Induce neuroregeneration	[100]
Spinal cord injury	HO1-ADMSC	Male SCI dog model	Gluteal subcutaneous fat from healthy beagle dog of age 1.5 years	Two injections into epicenter and one into the center of the injured segment	Alleviate inflammation	[100]
Spinal cord injury	Ngn2-ADMSCs	Female SCI rat model	Female Sprague–Dawley rats	Injection into the lesion epicenter	Differentiated into neurons, promote post-SCI functional recovery	[101]
Multiple sclerosis	IFN-β-ADMSCs	Female EAE mice model	Mice	Intravenous injection	Inhibit neuroinflammation	[102]
Multiple sclerosis	IFN-β-ADMSCs	Female EAE mice model	Inguinal fat of 6–8 weeks old C57BL/6 female mice	Intraperitoneal injection	Reduce IL-17; induce both IL-10 and Treg, alleviate cell infiltration	[103]
Multiple sclerosis	IL-4-ADMSCs	Female EAE mice model	Human ADMSC	Intraperitoneal injection	Reduce the clinical score of EAE	[104]
Parkinson’s disease	GDNF-ADMSC	6-OHDA-induced male SD rat model	Groin subcutaneous adipose tissue of Sprague-Dawley rat	Intrastriatal injection	Differentiate into dopaminergic neuron-like cells; clinical improvement	[105]
Parkinson’s disease	GDNF-ADMSC	Mild 6-OHDA hemiparkinson male rat model	Human adipose tissue	Intrastriatal injection	Immunomodulation and neuroprotection; restoration of tyrosine hydroxylase expressing cells	[106]

**Table 3 pharmaceutics-15-02637-t003:** Clinical trials of ADMSCs application in CNS disorders.

CNS Disorders	Status	Purpose	Enrollment	Intervention Model	Study Start Date	Study Completion Date	Main Outcomes	Reference/NCT Numbers
Stroke	Completed	To confirm the possible of autologous ADMSCs transplantation of chronic stroke	3 participants	Single-Group Assignment	19 October 2017	27 November 2018	All patients improved prominently, and no adverse effects or related safety issue detected within 6 months after treatment.	[110]; NCT02813512
Stroke	Active, not recruiting	To explore the dose of autologous ADMSCs transplantation	15 participants	Single-Group Assignment	6 February 2020	30 September 2023	NA	NCT04088149
Stroke	Recruiting	To investigate the safety, tolerability and efficiency of ADMSCs transplantation.	95 participants	Sequential Assignment	2 January 2018	1 July 2027	NA	NCT03570450
Traumatic brain injury	Active, not recruiting	To provide primary assessments of the safety, tolerability, and clinical effect of intravenous ADMSCs infusion in post concussion syndrome (PCS).	20 participants	Parallel Assignment	1 February 2021	31 January 2024	NA	NCT04744051
Spinal cord injury	Completed	To assess the clinical effects, change of quality of life, and mental signs after autologous ADMSCs transplantation to acute SCI.	47 participants	Parallel Assignment	NA	NA	Autologous ADMSCs transplantation was safe and provided both functional and post-SCI mental improvement.	[60]
Alzheimer’s disease	Recruiting	To evaluate the safety, severe adverse events, tolerability, and therapeutic effect of autologous ADMSCs intracerebroventricular injections in AD patients.	18 participants	Sequential Assignment	14 August 2023	March 2025	NA	NCT05667649
Multiple Sclerosis	Not yet recruiting	To investigate the safety and efficacy of escalating doses of ADMSCs SCM-010 in patients with secondary progressive multiple sclerosis (SPMS).	12 participants	Sequential Assignment	1 February 2023	1 February 2024	NA	NCT03696485
Amyotrophic Lateral Sclerosis	Completed	To evaluate the safety and possible efficiency of ADMSCs treatment in an ALS patient	1 participant	Single-Group Assignment	January 2015	January 2016	NA	NCT02383654

NA: The data is not applicable at the time of manuscript writing because it has not been published.

## Data Availability

Not applicable.

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
