# Peer review of "Advanced Progress in the Role of Adipose-Derived Mesenchymal Stromal/Stem Cells in the Application of Central Nervous System Disorders"

_pharmaceutics, 2023, doi:10.3390/pharmaceutics15112637_

Round 1
Reviewer 1 Report
Comments and Suggestions for Authors
H Wu's review article. Et al, submitted to pharmaceutcis “Advanced Progress in the Role of Adipose-derived Mesenchymal Stromal/stem Cells in the Application of Central Nervous System Disorders”. The objective of this systematic review addresses the importance of the role of adipose-derived Mesenchymal Stromal/ stem cells in the treatment of central nervous system disorders, exploring their therapeutic potential for clinical application.
- As it is mentioned in the manuscript that it is a review with a systematic summary, it needs to indicate how the articles in this review were chosen.
- I consider that the items covered in neurodegenerative diseases could be summarized in tables indicating the summary as cellular origin, sources, type of disease addressed, summarized experimental design and main outcomes.
- In clinical trials, numbers of registered studies could be indicated, in the same way that in pre-clinical studies tables would need to be created and show the main characteristics of the studies with their main results.
- I consider that the approach taken in the review does not provide a systematic approach and that the data shown in the way it was presented does not make the reader clear about the relevance of this cell type for ADSC. I believe that a significant change would need to be made to this manuscript.
Comments on the Quality of English LanguageIt´s ok
Reviewer 2 Report
Comments and Suggestions for Authors
the manuscript is very interesting, with an up to date review of the use of adipose-derived messenchimal stem cells or storm cells or exosomes derived from them to be used in regenerative therapy on the central nervous system. The review is very well structured, indicating the data available for different diseases from the central nervous system.
Very comprehensive and clear review.
Reviewer 3 Report
Comments and Suggestions for Authors
Dear colleagues!
Generally, the review is a concise and comprehensive work elaborating on a widely covered yet regularly updated field.
Improvements are required in 2 major points:
1) ADSC is a consistently outdated abbreviation for adipose-derived MSC (standing for mesenchymal stromal cells as suggested by A. Caplan in his Cytotherapy paper on nomenclature in 2017) so one may feel that terminology might be updated. Furthermore, introduction focuses on MSC stemness properties which in case of CNS have neglectible importance due to mainly paracrine mode of action. Thus, this might be a misleading rationale.
2) Little is provided on genetically modified adipose MSC (e.g. by viral vectors to express BDNF, GDNF etc.) which a were a major point during the last decade so this might be an important and relevant addition to the Review.
3) Some data on most important studies deserves a tabular presentation for Reader's convenience especially for clinical data.
4) Probably a separate passage on MSC secretome as an independent route of advances in the field is warranted (known as "cell therapy without cells")
Regards, Reviewer
Round 2
Reviewer 1 Report
Comments and Suggestions for Authors
Changes were made as requested.
Comments on the Quality of English LanguageMinor English language editing required
Reviewer 3 Report
Comments and Suggestions for Authors
Dear colleagues!
All queries were met in a sound manner.
Best, Reviewer